# Epigenetics in Obesity and Diabetes Mellitus: New Insights

**DOI:** 10.3390/nu15040811

**Published:** 2023-02-04

**Authors:** Rosario Suárez, Sebastián P. Chapela, Ludwig Álvarez-Córdova, Estefanía Bautista-Valarezo, Yoredy Sarmiento-Andrade, Ludovica Verde, Evelyn Frias-Toral, Gerardo Sarno

**Affiliations:** 1School of Medicine, Universidad Técnica Particular de Loja, Calle París, San Cayetano Alto, Loja 110101, Ecuador; 2Departamento de Bioquímica Humana, Facultad de Medicina, Universidad de Buenos Aires, Buenos Aires C1121ABE, Argentina; 3Hospital Británico de Buenos Aires, Equipo de Soporte Nutricional, Buenos Aires C1280AEB, Argentina; 4School of Medicine, Universidad Católica Santiago de Guayaquil, Av. Pdte. Carlos Julio Arosemena Tola, Guayaquil 090615, Ecuador; 5Carrera de Nutrición y Dietética, Facultad de Ciencias Médicas, Universidad Católica De Santiago de Guayaquil, Av. Pdte. Carlos Julio Arosemena Tola, Guayaquil 090615, Ecuador; 6Centro Italiano per la Cura e il Benessere del Paziente con Obesità (C.I.B.O), Department of Clinical Medicine and Surgery, Endocrinology Unit, University Medical School of Naples, Via Sergio Pansini 5, 80131 Naples, Italy; 7“San Giovanni di Dio e Ruggi D’Aragona” University Hospital, Scuola Medica Salernitana, 84131 Salerno, Italy

**Keywords:** obesity, diabetes, epigenetics, retinopathy, nephropathy, neuropathy

## Abstract

A long-term complication of obesity is the development of type 2 diabetes (T2D). Patients with T2D have been described as having epigenetic modifications. Epigenetics is the post-transcriptional modification of DNA or associated factors containing genetic information. These environmentally-influenced modifications, maintained during cell division, cause stable changes in gene expression. Epigenetic modifications of T2D are DNA methylation, acetylation, ubiquitylation, SUMOylation, and phosphorylation at the lysine residue at the amino terminus of histones, affecting DNA, histones, and non-coding RNA. DNA methylation has been shown in pancreatic islets, adipose tissue, skeletal muscle, and the liver. Furthermore, epigenetic changes have been observed in chronic complications of T2D, such as diabetic nephropathy, diabetic retinopathy, and diabetic neuropathy. Recently, a new drug has been developed which acts on bromodomains and extraterminal (BET) domain proteins, which operate like epigenetic readers and communicate with chromatin to make DNA accessible for transcription by inhibiting them. This drug (apabetalone) is being studied to prevent major adverse cardiovascular events in people with T2D, low HDL cholesterol, chronic kidney failure, and recent coronary events. This review aims to describe the relationship between obesity, long-term complications such as T2D, and epigenetic modifications and their possible treatments.

## 1. Introduction

The current view of epigenetics considers DNA modifications or associated factors with information content other than the DNA sequence itself. These modifications are maintained during cell division, influenced by the environment, and cause stable changes in gene expression. Hence, the epigenetic definition is now contemplated more dynamically than when Conrad Waddington first described it as the science that analyzes the link between genes and the events that give birth to the phenotype [1]. Acknowledgment of the profound effect of the environment on developmental plasticity, particularly with aging and susceptibility to common diseases, has brought major changes in epigenetics [2]. Some epigenetic DNA modifications that affect gene transcription are at least partially reversible, which may be enzymatically reversed after cessation of exposure to environmental factors, but some epigenetic changes persist over time [3].

The first hypothesis suggesting the role of epigenetics in the development of metabolic diseases described how early life events affect health outcomes in adulthood. However, this field of knowledge has reached enormous relevance with new laboratory technologies, especially with regard to epigenetic marks [4]. The significant epigenetic changes are constituted by DNA methylation, histone modifications (acetylation is the most known, but there are other mechanisms such as phosphorylation, ribosylation, ubiquitylation, SUMOylation, and citrullination) [5], and non-coding RNAs. In contrast, DNA and histone modifications close or open the chromatin structure regulating access to the transcription factors, and non-coding RNAs control gene expression at the RNA level [6]. DNA methylation, known as CpG (cytosine and guanine separated by only one phosphate group) methylation, silences gene expression, condensing chromatin [7]. Conversely, depending on the site of methylation and the number of methyl groups, histone methylation can either increase or decrease gene transcription, and acetylation relaxes the chromatin structure, allowing access to transcription factors for gene transcription [8,9]. Thus, histone acetylation is generally associated with active gene expression [8,9]. Non-coding RNAs are not considered epigenetic components; however, they are involved in epigenetic modifications [4,10,11]. MicroRNAs (miRNAs), for example, are one of the more abundant classes of gene regulatory molecules, and their dysregulation is associated with several diseases, including cancer and T2D [12]. Long non-coding RNAs (LncRNAs) regulate metabolic processes, and recent research has found that they play a determinant role in controlling pathophysiological processes in many chronic diseases, including cancer and T2D, and their complications [6].

Genetic and external factors are then involved in many diseases, and, in some cases, environmental changes result in adaptive epigenetic changes. These changes are heritable and can be altered in response to developmental signals or environmental exposures associated with adverse health conditions, although it is challenging to establish a causal role [3,13]. Some of these exposures have been studied under different conditions, such as maternal caloric restriction (starvation), caloric excess (diabetes) during gestation, and their association with obesity in offspring. Other conditions include ingesting endocrine-disrupting agents such as pesticides or plastics related to reproductive health and certain hormone-dependent cancers, inhalation of smoke and air pollutants with respiratory illnesses, and possible effects of stressful experiences [3,14,15]. Tumorigenesis is also partially regulated by epigenetic phenomena such as nucleosome remodeling by histone modifications, DNA methylation, and miRNA-mediated targeting of various genes [16].

In the same way, there is evidence that alterations in DNA methylation can contribute to the increased prevalence of obesity and both type 1 (T1D) and T2D [16]. Visceral fat (VF) is known to be associated with prediabetes and insulin resistance [17], and lipids secreted from obese adipose tissue accumulate in peripheral tissues such as the liver, pancreas, and muscle and alter insulin sensitivity causing T2D [18]. Studies included in a 2019 review have reported an epigenetic association with obesity, particularly with altered BMI and waist circumference, reporting differential DNA methylation and increased epigenetic variability in obese people [19]. In T2D, the development of micro- and macrovascular complications is correlated with DNA methylation, histone modifications, and several non-coding RNAs, such as miRNAs and LncRNAs, that affect many biochemical pathways. As these phenomena are being explored as therapeutic targets in the cancer field, they have also opened up the possibility of using them to treat T2D and prevent or slow down its complications [8,20,21,22,23]. This review is focused on various epigenetic factors, their involvement in the development of obesity and T2D, and the prescribed drugs in altering those mechanisms.

## 2. Obesity, Inflammation, and Epigenetics

Patients with obesity have been shown to have epigenetic alterations associated with CpG methylation and the expression pattern of miRNAs at all stages of adipocyte differentiation, affecting their lipid metabolism and insulin sensitivity in adipose tissue and causing β-cell dysfunction [23,24]. These epigenetic alterations persist even without causal stimuli (high-fat diet and sedentary lifestyle), leading to persistent inflammation, and are called metabolic memory. In turn, this causes vascular damage and the development of cardiovascular disease (CVD), affecting the immune system of patients with obesity and favoring infections, morbidity, and mortality [25].

Variabilities in DNA methylation at the HIF3A locus have been reported in the blood cells and adipose tissue of people with obesity [26]. Reprogramming of DNA methylation in PPARGC1A (the gene encoding PGC1, a master regulator of biogenesis and mitochondrial function) has also been shown in subjects with obesity of all ages, as well as hypermethylation of the proopiomelanocortin (POMC) promoter of intron 2/exon 3 in the melanocortin system. These results imply that the development of obesity depends critically on mitochondrial function [27,28].

As described above, histone modifications are another epigenetic process linked to obesity. Among the histones related to TNF-α and IL6 is H3K4, and these proinflammatory cytokines are involved in insulin signaling [29]. miRNAs are regulators of adipogenesis, which involves processes such as hyperglycemia or insulin regulation. The secretion of CCL2 (a gene linked to proinflammatory cytokines such as TNF-α and IL-1 β) is influenced by approximately 20 miRNAs. The low regulation of this gene produces adipose inflammation that increases the secretion of proinflammatory cytokines such as TNF-α and IL1β present in the pathophysiology of T2D [29] (Figure 1).

The permanence of genetic and environmental factors, such as a high-fat diet, a sedentary lifestyle, or toxic agent exposure, leads to chronic metabolic inflammation, which can cause hyperglycemia, IR, β-cell dysfunction, and as a consequence, T2D [30,31]. Obesity is known as a promoter of T2D. One of the best-proven hypotheses is the chronic inflammatory mechanism in people with obesity that interferes with insulin signaling, or IR processes. Epigenetic mechanisms control the expression of proinflammatory cytokines such as tumor necrosis factor-alpha (TNF-α), interleukins such as IL6, or cells such as monocytes and granulocytes [31,32].

Several genes are involved in the presence of T2D and are associated with obesity, as shown by the GWAS studies (ADCY5, FTO, HHEX, IRS1, KCNQ1, PPARG, and TCF7L2) [33]. Among the epigenetic processes involved in DNA methylation in patients with obesity, they were seen to be mainly mediated by DNA methyltransferase DNMT3A. Together with the Fabp4 promoter, it increased the expression of proinflammatory cytokines linked to M1 macrophages (IL-6 and TNF-α). Moreover, the methylation of SCD1 and SLCA1 is also related to inflammatory processes with increased cytokine production. Moreover, the inhibition of anti-inflammatory processes and increased secretion of proinflammatory cytokines are linked to DNMT1 methylation [34,35]. In addition, hypoxia or periconceptional malnutrition modulate epigenetic processes such as DNA methylation in genes such as HIF in hypoxia and LEP, INSIGF, and GNASAs in periconceptional malnutrition, promoting metabolic or inflammatory processes that lead to obesity or diseases such as T2D [36].

## 3. Epigenetics in Diabetes Mellitus

The incidence of T2D in the world is proliferating, with it becoming a public health problem that represents enormous costs for individuals who suffer from it and society [37,38]. The need to extensively know the pathophysiology and, therefore, find preventive and treatment measures has led to the search for different processes that allow us to understand the disease better, one of them being epigenetics. The clinical manifestations of T2D induced by alterations in gene expression that are not associated with specific changes in the DNA sequence [10] are produced by the interaction between genetic and environmental factors. Among the latter are intrauterine environments, stress, inadequate nutrient intake, low birth weight, older age [10,39], physical inactivity, diet, and obesity [4]. The last one is the most critical factor for T2D. As a result of these alterations, changes in glucose–insulin metabolism may occur, including impaired pancreatic β-cell function and liver glucose metabolism [40]. These disturbances lead to subsequent insulin insufficiency and resistance in the target organs (the liver, muscle, and adipose tissue) [41], which exhibit inappropriate expression of profibrotic and proapoptotic genes [10,40]. It has been suggested that DNA methylation may be a reversible and dynamic process that requires continuous regulation, and modifications have been described in the methylation of some genes of the insulin pathway, which has provided information on the possible relevant role of epigenetics as a key factor in positive changes in metabolic control parameters in T2DM patients [42,43].

Previously it was mentioned that the main mechanisms involved in the epigenetic modification of T2D are histone modifications, DNA methylation/demethylation within CpG islands, and non-coding RNA [10]. A vital point is the interrelationship between DNA methylation and histone modification, which influence gene transcription factors [40]. DNA methylation can alter the expression of specific genes. CpG islands play an essential role in this process. DNA methylation occurs primarily at CpG islands that impact gene expression (promoting gene silencing), cell differentiation, and molecular response. In a case–control study of Asian Indian and European T2D patients to determine whether DNA methylation is associated with the incidence of T2D, it was shown that methylation markers in seven genetic regions (TXNIP, PROC, C7orF29, SREBF1, PHOSPHO1, COCS3, and ABCG1) were associated with a higher incidence of T2D and four times higher risk of presenting the pathology in cases versus controls [24].

DNA methylation also causes alterations in pancreatic islets, adipose tissue, skeletal muscle, and the liver [40]. DNA methylation was identified at the TCF7L2, KCNQ1, THADA, FTO, IRS1, and PPARG loci in islets from T2D donors [33], and some of them were associated with altered expressions of mRNA, implying that illness development is linked to a change in transcriptional activity [33]. Other DNA-methylation-associated alterations include the pancreatic and duodenal homeobox 1 (PDX1), TCF7L2 genes, the peroxisome proliferator-activated receptor gamma coactivator 1-alpha (PPARGC1A), PAX4 genes, and the glucagon-like peptide-1 receptor (GLP1R) gene that deregulate insulin secretion in β cells and therefore reduce expression in pancreatic islets; in addition to nuclear factor-kappa β (NF-kβ), osteopontin and Toll-like receptors that generate proinflammatory signals [33].

In 2013, a complete genome sequencing identified 25,820 differentially methylated regions (DMRs) covering the promoters of PDX1, TCF7L2, and adenylate cyclase 5 (ADCY5) [44]. In 2014, genome-wide methylation quantitative trait locus (metQTL) analysis included genotype data for 574,553 single nucleotide polymorphisms (SNPs) with DNA methylation. Data were extracted for 468,787 genes, and pancreatic islet CpG sites from 89 donors showed that SNPs influence gene methylation (adenylate cyclase 2 (ADCY2), potassium inwardly rectifying channel subfamily J member 11 (KCNJ11), traits of glucose-like growth factor receptor bound protein 10 (GRB10), and PDX1), all of which are responsible for epigenetic deregulation in pancreatic islets of T2D patients [45]. Recently, the analysis of 27,578 CpG sites from 14,495 genes from obese people with T2D detected a more significant amount of DMRs in visceral adipose tissues (340 DMRs), and the liver (185 DMRs), which favors insulin resistance (IR), adipogenesis, fat storage, and pro-inflammation in these obese patients [46]. Later, hypermethylation of nuclear receptor subfamily 4 group A member 1 (NR4A1), a transcriptional regulator of glucose metabolism in the skeletal muscle and liver, was observed in blood samples from diabetic patients [47].

Acetylation and methylation at the amino-terminal lysine residue is the primary mechanism of epigenetic modification of histones in muscle, the clinical result of which leads to IR being expressed in two ways: in peripheral use (GLUT 4 alteration) and hepatic noninhibition of gluconeogenesis; there is hepatic resistance to inhibit this last process [10]. The consequence is an increase in fasting blood glucose that causes greater stimulation of the islet [48]. GLUT-4 is an insulin-regulated glucose transporter found in adipocytes, skeletal muscle, the liver, and myocardium, which favors glucose’s movement into the cell. Epigenetic modifications can decrease the expression of this transporter and, as a consequence, glucose transport [10,48].

The stability of gene expression occurs due to the balance between the histone acetylation and deacetylation processes without affecting DNA sequencing [10,49]. Histone modification is associated with pancreatic β-cell death due to H3K9 deacetylation that inactivates chromatin, leading to the silencing of the PDX1 gene [10,49]. At the muscle level, H3K9 deacetylation produces peripheral inactivation of the GLUT 4 transporter, leading to hyperinsulinemia and hyperglycemia [10,49].

Another mechanism that regulates gene expression at both the transcriptional and post-transcriptional levels is ncRNA [50]. miRNAs are ncRNAs that bind with messenger RNA to regulate physiological and pathological events [15,50,51]. Currently, miRNAs act on target cells responsible for hepatic glycogen metabolism and insulin secretion, causing destruction and apoptosis of β cells and IR in peripheral organs [50]. The following miRNAs give these metabolic disorders: let-7b, miR-103, miR-142, miR-144, miR-223, miR-29, and the miR124a and miR-375 genes that have been considered to have a higher risk of causing T2D, altered fasting glucose, and IR [40,51,52]. Furthermore, exosomes that carry miRNAs contribute to the progression and complication of T2D. These epigenomic processes are also regulated by DNA methylation, histone modification, and vice versa [50,53,54].

## 4. From the Bench to the Bedside

The importance of intensive glucose control in diabetic patients is challenging. Complications in this pathology involve microangiopathic, macroangiopathic, musculoskeletal, neurological, and other events [55]. In this chapter, the epigenetic changes of chronic complications of diabetes mellitus (DM) focusing on T2D will be reviewed (Table 1). In addition, epidemiologic data on epigenetic changes will be reviewed.

A multicenter clinical trial enrolled 500 participants with a mean age of 26.4 +/− 2.8 years and a diagnosis of youth-onset T2D 13.3 +/− 1.8 years ago. This observational follow-up study (performed from 2011 to 2020) was conducted in two phases; at the end of the second phase of follow-up, the trial pointed out that the risk of micro- and macrovascular complications increased over time and that the risk factors for the development of such complications were hyperglycemia, minority race, hypertension, and dyslipidemia [56].

In 499 participants in the Diabetes Control and Complications Trial (DCCT) and the Epidemiology of Diabetes Interventions and Complications (EDIC) follow-up study using blood DNA, the association between methylation of DNA in CpG and HbAb1 was demonstrated. These results suggested that DNA methylation in CpG may play an essential role in hyperglycemia and complications caused by metabolic memory (it negatively increases long-term complications if there was a previous period of hyperglycemia) [57]. In another cohort study in a rural Chinese population with 286 cases of T2D and matched controls, they found an increase of 16% (OR = 1.16, 95% CI = 1.02–1.31) in the incidence of T2D. These findings were identified when detecting an increase in DNA methylation levels at the ABCG1, CpG13, and CpG14 loci. Similarly, this risk increased to 78% when there was a gain in DNA methylation at the CpG15 locus, suggesting an etiological model for T2D [58].

### 4.1. Diabetic Nephropathy and Epigenetics

Microangiopathies result from the thickening of the basement membrane and glycosylation of structures resulting from T2D. Diabetic nephropathies (DNe) are associated with glomerulosclerosis, caused by an inflammatory process secondary to hyperglycemia. In addition, it is characterized by the presence of albuminuria and a decrease in the glomerular filtration rate generated by apoptosis and functional loss or alteration of podocytes, endothelial cells, and mesangial cells [59,60].

DNA methylation in cytosine in the CpG island is present in renal damage in diabetic patients. This methylation process is mediated by DNMT1 and DNMT3a/3b [61]. In an exploratory study of 107 cases, 253 controls, and 14 experimental controls, several genes were associated with end-stage kidney disease in T1D patients (AFF3, ARID5B, CUX1, ELMO1, FKBP5, HDAC4, ITGAL, LY9, PIM1, RUNX3, SEPTING9, and UTF3A) [62].

Histone modification is another process that plays an essential role in the epigenetics of DNe and metabolic memory. These processes are mediated by hyperglycemia and in response to the presence of transforming growth factor-beta (TGF-β), triggering active methylation of histones such as H3K4, H3K27, and H3K9 [63,64].

Multiple miRNAs regulate the characteristics of DNe. In a study carried out on 68 patients with T2D and 11 healthy subjects, whose objective was to establish a relationship between podocyte damage and proximal tubule dysfunction, it was shown that there is an association between podocyte damage and impaired proximal tubule function and excreted miRNA. Among the multiple regression models that showed correlation were miRNA192 with synaptopodin, uNAG, and eGFR (R^2^ = 0.902; *p* < 0.0001), miRNA124 with synaptopodin, uNAG, UACR, and eGFR (R^2^ = 0.881; *p* < 0.0001), and miRNA21 with podocalyxin, uNAG, UACR, and eGFR (R^2^ = 0.882; *p* < 0.0001) [65]. Together with the action of miRNAs, LncRNA is a group of transcription materials that lack a complete open reading frame and have no protein-coding functions [66]. A cross-sectional study of 136 participants with T2D and 25 healthy subjects demonstrated that LncRNAs contribute to the pathogenesis of DNe, with LncRNA MALAT1 and LncRNA NEAT1 through miRNA methylation associated with biomarkers of proximal tubule and podocyte injury [66].

Another factor related to the severity and presence of DNe is S-adenosylhomocysteine (SAH), a product of DNA methylation and histone modification [67]. Although the pathophysiological process of nephron damage is unknown, a study carried out in C57BL/6 mice indicated that inhibiting S-adenosylhomocysteine hydrolase (SAHH) induces an increase in SAH, allowing for the accumulation of SAH involved in damage at the podocyte level (renal function). In addition, the rise in SAH deters H3N27 me3 and increases the early growth response promoter (EGR1) [67]. These changes lead to oxidative stress due to the activation of the protein that interacts with thioredoxin (TXNIP) and NLPR3 (a mediator of the innate immune system), thus aggravating DNe [67].

### 4.2. Diabetic Retinopathy and Epigenetics

Diabetic retinopathy (DR) is another microvascular alteration in patients with T2D, whose hyperglycemic state causes progressive and irreversible blindness, even in normoglycemic diabetic states [68]. In addition, their hyperglycemic state causes modifications in DNA methylation in the early stages of T2D due to the metabolic memory present in this condition [69]. Genome-wide studies identified that the vascular changes of DR are regulated by LncRNAs characterized by pathological proliferation of retinal vessels, thickening of the capillary basement membrane associated with increased vascular permeability, tissue ischemia, and release of vasoactive substances [70].

Three situations may occur: (a) retinal neovascularization with vitreous hemorrhage and retinal detachment, called proliferative DR; (b) a process without neovascularization called non-proliferative DR with the formation of microaneurysms and subtle dilation of retinal vessels, generally in the early stages; and (c) neurodegeneration that causes neuroapoptosis, resulting in the death of retinal ganglion cells, glial alteration, and abnormalities in the retinal pigment layer [71].

Recent research on the molecular mechanisms of epigenetics has revealed that oxidative stress is the promoter of these alterations in photoreceptor cells since they are rich in easily oxidized polyunsaturated fatty acids. Therefore, the epigenetic changes caused by hyperglycemia lead to an excessive accumulation of cytosolic reactive oxygen species (ROS) [72] due to the activation of Nox2 (NADPH oxidase 2 nicotinamide adenine dinucleotide phosphate) [73].

Five established pathways produce metabolic abnormalities due to oxidative damage: the increased flow of the polyol pathway (sorbitol–aldose reductase) and the hexosamine pathway, hyperactivation of protein kinase C isoforms and angiotensin II, and the concentration of advanced glycation end products (AGEs) [71,74]. In addition, the polyol pathway has been investigated as a pathogenic role of DR. This pathway is activated in endothelial cells, pericytes, and Müller cells to produce macular edema and subsequent retinal ischemia [75,76]. Wenliang Li et al. investigated 3000 subjects, 1500 patients without DR and 1500 with it, of which 750 had non-proliferative DR and 750 had proliferative DR. They found that ALR2 rs759853 polymorphism variants were significantly associated with an increased risk of DR, *p* < 0.01 [77].

DR patients may have hyperlipidemia associated with hyperglycemia that further increases Rac1–Nox2–ROS activation, causing progressive mitochondrial damage with the loss of capillary cells [73], cell apoptosis, inflammation, lipid peroxidation, and structural and functional alterations in the retina [71]. A recent study compared retinal epigenetic modifications at 2, 4, and 6 months in non-obese T1D-induced rats versus obese T2D rats. It concluded that obesity accelerates mitochondrial damage by causing a more significant accumulation of cytosolic ROS by Rac1–Nox2 signaling, *p* = <0.05, establishing DR in the short term [73].

### 4.3. Diabetic Neuropathy and Epigenetics

Diabetic neuropathy is a very complex complication of T2D that can appear as diabetic peripheral neuropathy (DPN), diabetic autonomic neuropathy, or uremic neuropathy [78,79,80]. DPN is caused by chronic hyperglycemia, leading to peripheral nerve damage and subsequent foot ulceration, Charcot neuropathy, possible amputation, and death [79].

miRNAs also play a role in the pathogenesis of diabetic neuropathy as there are genes that are expressed in the brain and peripheral neuronal tissue [81]. In 2017, Ying-Bo Li et al. published a study that included 60 patients, of whom 75% had an increase in the expression of the mir-199a-3p gene with a significant association with a longer duration of the disease (*p* = 0.041) [82]. Subsequently, the study carried out by Cinzia et al. describes the genetic variation of mir-499a as a probable association with diabetic neuropathy [83]. On the contrary, Yanzhuo et al. identified the mir-25 gene as a protective factor for diabetic neuropathy [84]. Another work with diabetic rodents reports the association of miRNA-29c and protein kinase C gene expression with distal neural damage [85].

LncRNA has been considered the epigenetic mechanism that mediates this damage [79]. In 2018, Muhamad et al. evaluated the epigenetic mechanisms of the expression of LncRNA present in DPN in samples from 18 different participants, evenly distributed between DPN and healthy groups. There, 17 genes were found to be differentially expressed between both types of patients (*p* = <0.0005), two of which (MTHFSD and LMAN2L) were partially non-coding. In addition, the TUBA4B gene was overexpressed in the DPN samples; however, this study has inconsistent results [78], and, like other publications, the studies are limited due to the variability of epigenetic mechanisms in these patients [79].

## 5. Is There a Possible Therapeutic Related to Epigenetics?

Epigenetic factors and signal transduction pathways contribute significantly to DM; they affect gene expression and phenotypes associated with DNe. Life-threatening complications, including DNe and significant cardiovascular problems, denote the need for a more molecular mechanism to clarify better therapies for DM [86]. Furthermore, epigenetic modifications can lead to severe phenotype changes, and these events can be reversible [87]. Likewise, with environmental and genetic stress, epigenetic changes push ahead or backward, and stress removal can revert to the original state. Ongoing work is building to develop novel drug candidates to treat DM; they can reverse the phenotype changes, primarily when therapy has been administered early in the disorder’s progression [10]. New and novel T1D and T2D management focuses on epigenetic modifications for glucose metabolism, and some new small molecules demonstrate epigenetic activity [88]. According to their epigenetic effects, they are called “epidrugs” and are still under clinical trials. In the event of chromatin modification, they are referred to as histone acetyltransferase inhibitors (HDAC inhibitors) and RNA interference molecules [89].

Recently, data proposed a detailed mechanism related to modifying gene expression to manipulate the etiology of T2D and its complications. One of the essential issues of diabetic stimuli triggers some epigenetic modifications to be the basis of metabolic memory [90,91,92]. Nevertheless, there remains a lack of information regarding epigenetic factors and the mechanism related to gene modification; this is probably the rationality behind the rapid growth and energetic field for new mechanisms and therapeutic targets [93].

Hyperglycemia and its related complications are the main objectives for conventional management, including diet, insulin, and/or anti-diabetic drugs. Their correct management favors patients; however, these traditional medications have substantial side effects, such as lactic acidosis, severe hypoglycemia, cardiovascular problems, and urinary tract infections [94]. Although hypoglycemia is a fear side effect, its relationship with patient mortality is currently under discussion [94]. Epigenetic drugs might improve T2D complications; the critical issue is epigenetic modification [19].

Epigenetic modulators are a brand-new drug therapy that modifies gene transcription [89]; BET domain proteins operate like epigenetic readers and communicate with chromatin to access DNA for transcription. BET proteins attach molecular scaffolds using chromatin and transcription machinery to promote transcription and mRNA production and provide maladaptive gene expression in CVD models [95]. BET protein inhibitors can modify the disease-driven cellular response in people with a probability of CVD, together with CKD [96,97]. Apabetalone, an oral BET inhibitor, has anti-inflammatory and alkaline-phosphatase-lowering attributes [92,97,98].

The leading causes of CKD are T2D and hypertension, accounting for more than 50% of the cases [99]. DM and CKD are related to coronary and cerebrovascular disease, heart failure, and death [100]. Inflammation is extremely crucial in controlling the cardiovascular risk of CKD, regardless of T2D and its etiology [101]. Conventional treatment has reduced cardiovascular risk in patients with moderate CKD, but the residual risk remains considerable [102]. Apabetalone has been studied for preventing major adverse cardiovascular events (MACE) compared with novel treatment vs. a placebo in patients with T2D, low HDL cholesterol, and recent coronary events. Apabetalone performs well in lowering MACE, but the effect did not reach statistical significance [91,92]. However, in a phase 3 controlled post-acute coronary syndrome trial in patients with T2D, the participants with CKD had a better response, with an almost 50% reduction in MACE up to 27 months of inhibition. This randomized, double-blind, placebo-controlled, multicenter trial was conducted in 13 countries with 2425 participants. In the apabetalone group, more individuals abandoned the study than the placebo group (9% vs. 6%) for medical reasons [91].

The placebo’s main results summarized that patients with CKD presented with acute coronary syndrome and myocardial infarction; the proportion without ST elevation was more significant in the placebo group. The main laboratory findings for apabetalone treatment were a decrease in serum alkaline phosphatase compared to the placebo group, especially in participants with CKD, and a modest increase in HDL cholesterol [103]. Participants who completed the follow-up period of the placebo group with CKD underwent a higher incidence of MACE than those without CKD. Table 2 summarizes the trial results showing a significant reduction in primary HRs and Cis outcomes for indicating composite and component endpoints [103]. Although, in a post-hoc analysis of the BETonMACE trial, including patients with T2D, recent acute coronary disease, and a moderate likelihood of NAFLD, apabetalone reduced the risk of MACE [104].

Apabetalone was related to a declining hazard for MACE and fewer hospitalizations for heart failure [90]. Apabetalone trials in patients with T2D and CKD had a better response to the treatment, fewer hospitalizations, and a significant reduction in MACE [90]. This novel treatment is a promising trial of epigenetic modification with secure and efficient oral medication with inhibition of BET protein to reduce cardiovascular risk in individuals with CKD, T2D, and recent acute coronary syndrome. Apabetalone is considered a selective BET protein inhibitor with possibly healthy properties on pathways related to inflammation, thrombosis, and vascular calcification. Nevertheless, the mechanism underlying these benefits is still unclear [90,91].

## 6. Conclusions

Various epigenetic factors are involved in developing T2D, such as DNA methylation, which causes alterations in pancreatic islets, adipose tissue, skeletal muscle, and the liver. Many genes are implicated in deregulating insulin secretion in pancreatic β cells and inducing proinflammatory signals. Other epigenetic alterations in visceral adipose tissues and the liver favor IR. At the same time, acetylation and methylation of histones in muscle are expressed in two ways: in the peripheral insulin use and the hepatic noninhibition of gluconeogenesis, and, consequently, there is an increase in fasting blood glucose that causes greater stimulation of the islet.

Epigenetic modifications can also decrease the glucose transporter’s expression and alter glucose’s movement into the cell. Histone acetylation and deacetylation processes can also induce pancreatic β cell death or inactivation of transporters at the muscle level, causing hyperinsulinemia and hyperglycemia.

Currently, miRNAs act on the target cells responsible for hepatic glycogen metabolism and insulin secretion, causing destruction and apoptosis of β cells and IR in the peripheral organs. Therefore, the two general epigenetic modifications and ncRNA stated in this review have been described at different levels of glucose metabolism and IR, and so in T2D.

Some studies have found that epigenetic modifications in DM and microvascular complications that affect small blood vessels are also related. DNA methylation and histone modifications are processes of enormous significance in the epigenetics of DNe and metabolic memory. These processes are mediated by hyperglycemia. Multiple miRNAs and LncRNA regulate the characteristics of DNe, especially related to podocyte and proximal tubule injury. Research in animals has shown that SAH, a product of DNA methylation and histone modification, is related to the presence and severity of DNe. LncRNAs regulate vascular changes in DNE. On the other hand, DPN has been linked to miRNAs and LncRNA. These two epigenetic factors have been considered the mediating epigenetic mechanisms responsible for this complication of T2D.

Research in therapeutic advances in this area has been published with drugs that alter those mechanisms. In particular, in CVD and T2D, BET protein inhibitors are being studied in clinical trials due to their property to modify the disease-driven cellular response in people with the probability of CVD and CKD. The BD2-selective BET inhibitor apabetalone has been used in trials as a therapy for epigenetic changes, and it has shown that it might decrease cardiovascular risk in individuals with CKD, T2D, and recent acute coronary syndrome and also might modulate the pathways associated with inflammation, vascular calcification, and thrombosis, but the mechanism underlying this benefit is still unclear. Although the reported evidence between the association of complications related to obesity and diabetes and epigenetic modifications is unclear as to whether they are a cause or a consequence, it would seem that treatment with drugs that modulate said modifications could be a promising therapy. The results are inconsistent with subgroups of patients and post-hoc results, but much still needs to be explored in the future.

## Figures and Tables

**Figure 1 nutrients-15-00811-f001:**
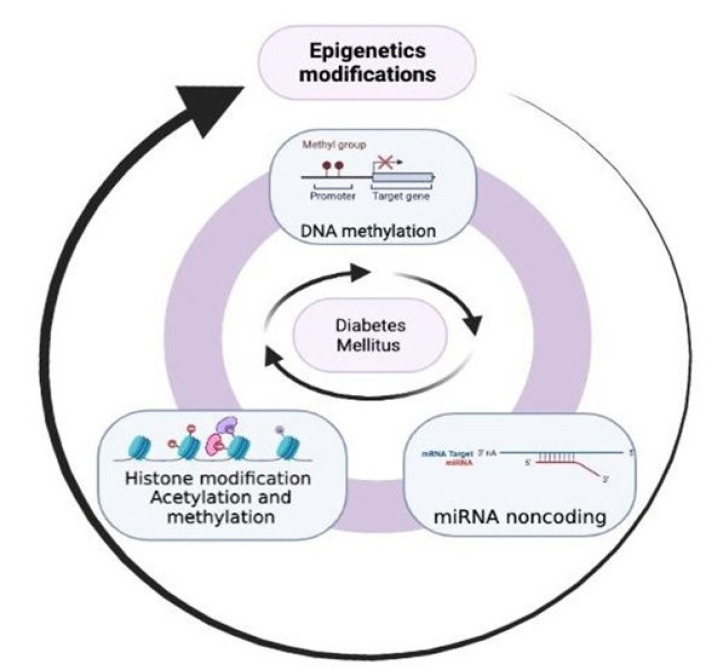
The role of epigenetics in the pathogenesis of type 2 diabetes. DNA methylation, histone modifications (acetylation/methylation), and noncoding RNA are present through different pathways of the pathophysiology of the disease.

**Table 1 nutrients-15-00811-t001:** Specific epigenetic modifications in T2D complications.

	Diabetic Nephropathy	Diabetic Retinopathy	Diabetic Neuropathy
DNA Methylation	AFF3, ARID5B, CUX1, ELMO1, FKBP5, HDAC4, ITGAL, LY9, PIM1, RUNX3, SEPTING9, UTF3A		
Histone acetylation/methylation	H3K4, H3K27, H3K9		
Non-coding RNA	miRNA192, miRNA124, miRNA21, LncRNA MALAT1, LncRNA NEAT1	LncRNA	mir-199a-3p, mir-499a, miRNA-29c, lncRNA MTHFSD, lncRNA, LMAN2L

Abbreviations: miRNA: microRNAs; LncRNA: long non-coding RNA.

**Table 2 nutrients-15-00811-t002:** Specific epigenetic modifications in T2D complications.

	eGRF < 60 mL/min per 1.73 m^2^		eGRF > 60 mL/min per 1.73 m^2^	
Variable	Placebo	Apabetalone	Placebo	Apabetalone
Significant reduction of primary outcome MACE	None	Positive	None	None
Composite events MACE and CHF	None	Positive	None	None
Components (# of events)				
CV death	10	5	4	4
Non-fatal MI	12	7	7	6
Non-fatal stroke	4	2	1	1
CHF hospitalization	9	2	3	2

Abbreviations: eGFR, glomerular filtration rate; MACE, mayor adverse cardiovascular events; CHF, congestive heart failure; CV, cardiovascular; MI, myocardial infarction.

## Data Availability

Not applicable.

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
