# Peer review of "Epigenetics in Obesity and Diabetes Mellitus: New Insights"

_nutrients, 2023, doi:10.3390/nu15040811_

Round 1
Reviewer 1 Report
This paper is aimed "to describe the relationship between obesity, long-term complications such as T2D, and epigenetic modifications and their possible treatments", but in its status it has limited perspectives to provide claimed "new insights" in this direction. Essentially, extensive literature and reviews reporting epigenetic specificities of diabetic conditions are outlined. There is a wide description of epigenetic changes occurring in numerous sets of patients and experimental systems, both referring to diabetes occurrence and of its complications. However, no possible causal pathogenetic processes are indicated to indicate potential targeting of therapies, in particular with apabetalone. While certainly manuscript indicates that the use of this molecule deserves further research to assess how its efficacy can be exerted through modulations of epigenetic mechanisms, not other insights about diabetes pathogenesis can be inferred to be possibly used to improve diagnosis and therapy of this disease. The potential activity of apabetalone within obesity and diabetes pathogenetic mechanisms is just an hypothesis. while available data still only indicate its efficacy vs. the cardiovascular complications of these diseases. Therefore manuscript requires an extensive revision to evidence which reported epigenetic mechanisms can be causal and not consequential within obesity and diabetes pathogenesis and which of them could be included among those affected by apabetalone.
Author Response
This paper is aimed "to describe the relationship between obesity, long-term complications such as T2D, and epigenetic modifications and their possible treatments", but in its status it has limited perspectives to provide claimed "new insights" in this direction. Essentially, extensive literature and reviews reporting epigenetic specificities of diabetic conditions are outlined. There is a wide description of epigenetic changes occurring in numerous sets of patients and experimental systems, both referring to diabetes occurrence and of its complications.
Author´s response: First of all, we would like to thank the reviewer for the job he did reading and making recommendations. These were very fruitful and have improved the quality of the work. . All the corrections have been marked up using the “Track Changes” function from MS Word in the revised version for the Reviewer's convenience.
The work has undergone many changes, with citation updates and a greater focus on T2D. The title has been changed and part of the manuscript has been restructured.
However, no possible causal pathogenetic processes are indicated to indicate potential targeting of therapies, in particular with apabetalone. While certainly manuscript indicates that the use of this molecule deserves further research to assess how its efficacy can be exerted through modulations of epigenetic mechanisms, not other insights about diabetes pathogenesis can be inferred to be possibly used to improve diagnosis and therapy of this disease. The potential activity of apabetalone within obesity and diabetes pathogenetic mechanisms is just an hypothesis. while available data still only indicate its efficacy vs. the cardiovascular complications of these diseases.
Author´s response: Thank you for the comment. We agree that the potential activity of apabetalone in diabetes may be a hypothesis. We added a paragraph to clarify this position in lines 469-473:
“Although the reported evidence between the association of complications related to obesity and diabetes and epigenetic modifications is not clear whether they are a cause or a consequence, it would seem that treatment with drugs that modulate said modifications could be a promising therapy. The results to date are inconsistent, with subgroups of patients and with post hoc results, but much remains to be explored in the future”
Therefore manuscript requires an extensive revision to evidence which reported epigenetic mechanisms can be causal and not consequential within obesity and diabetes pathogenesis and which of them could be included among those affected by apabetalone
Author´s response: The drug was not used in obese patients, it was used in patients with T2D who could occasionally be obese.

Reviewer 2 Report
The authors have written a comprehensive review on the role of epigenetics in obesity and diabetes. Overall, they have provided a detailed look at our current understanding of this topic. They have included a section of therapeutics targeting epigenetics.
The main concern for this reviewer is the large number of review articles they cite. There are more reviews cited than actual research articles. The original research papers should be cited when discussing data.
Other minor concerns include:
The title Is confusing – epigenetics is not a comorbidity for obesity and diabetes.
There is repetition between the different sections, which makes it cumbersome.
The division of the material into the different sections is confusing. Section 2 (Obesity, Inflammation, Epigenetics, and Diabetes) and Section 3 (Epigenetics in Diabetes mellitus, physiopathology) are overlapping. The authors should focus on obesity, inflammation, and epigenetics in Section 2 and on diabetes and epigenetics in Section 3.
Lines 112 and 113 – the POMC gene is not expressed in adipose tissue.
There are several errors in grammar or sentence construction.
Author Response
The authors have written a comprehensive review on the role of epigenetics in obesity and diabetes. Overall, they have provided a detailed look at our current understanding of this topic. They have included a section of therapeutics targeting epigenetics.
Author´s response: First of all, we would like to thank the reviewer for the job he did reading and making recommendations. These were very fruitful and have improved the quality of the work. All the corrections have been marked up using the “Track Changes” function from MS Word in the revised version for the Reviewer's convenience.
The main concern for this reviewer is the large number of review articles they cite. There are more reviews cited than actual research articles. The original research papers should be cited when discussing data.
Author´s response: We appreciate the reviewer’s observations. The work has undergone many changes, with citation updates with primary resources, not reviews. Also, the article has greater focus on T2D.
Other minor concerns include:
The title Is confusing – epigenetics is not a comorbidity for obesity and diabetes.
Author´s response: We appreciate the reviewer’s observations. The title has been changed and part of the manuscript has been restructured.
There is repetition between the different sections, which makes it cumbersome.
The division of the material into the different sections is confusing. Section 2 (Obesity, Inflammation, Epigenetics, and Diabetes) and Section 3 (Epigenetics in Diabetes mellitus, physiopathology) are overlapping. The authors should focus on obesity, inflammation, and epigenetics in Section 2 and on diabetes and epigenetics in Section 3.
Author´s response: We thank the reviewer’s suggestion and modify structure of the text as follows:
- Introduction
- Obesity, Inflammation and Epigenetics
- Epigenetics in Diabetes mellitus
- From the bench to the bedside
- Diabetic nephropathy and epigenetics
- Diabetic retinopathy and epigenetics
- Diabetic neuropathy and epigenetics
- Is there a possible therapeutic related to epigenetics?
- Is there a possible therapeutic related to epigenetics?
Lines 112 and 113 – the POMC gene is not expressed in adipose tissue.
Author's response: We thank the reviewer for the suggestion. That sentence has been corrected as follows:
Lines 122-127: A reprogramming of DNA methylation in PPARGC1A (the gene encoding PGC1, a master regulator of biogenesis and mitochondrial function) has also been shown in obese subjects of all ages, as well as hypermethylation of the proopiomelanocortin (POMC) promoter of intron 2 / exon 3 in the melanocortin system. These results imply that the development of obesity depends critically on mitochondrial function
There are several errors in grammar or sentence construction.
Author’s response: We appreciate the reviewer’s observations. For the grammar and spelling mistakes, the paper was reviewed by a native English speaker.
Round 2
Reviewer 1 Report
Present form is acceptable